# “All for One and One for All”: The Secreted Heat Shock Protein gp96-Ig Based Vaccines

**DOI:** 10.3390/cells13010072

**Published:** 2023-12-29

**Authors:** Laura Padula, Eva Fisher, Natasa Strbo

**Affiliations:** Department of Microbiology and Immunology, Miller School of Medicine, University of Miami, Miami, FL 33136, USA; lromero@med.miami.edu (L.P.); efisher@med.miami.edu (E.F.)

**Keywords:** heat shock proteins, gp96, vaccine, antigen cross presentation, mucosal immunity, innate and adaptive immune responses

## Abstract

It has been 50 years since Peter Charles Doherty and Rolf M Zinkernagel proposed the principle of “simultaneous dual recognition”, according to which adaptive immune cells recognized “self” and “non-self” simultaneously to establish immunological efficacy. These two scientists shared the 1996 Nobel Prize in Physiology or Medicine for this discovery. Their basic immunological principle became the foundation for the development of numerous vaccine approaches against infectious diseases and tumors, including promising strategies grounded on the use of recombinant gp96-Ig developed by our lab over the last two decades. In this review, we will highlight three major principles of the gp96-Ig vaccine strategy: (1) presentation of pathogenic antigens to T cells (specificity); (2) activation of innate immune responses (adjuvanticity); (3) priming of T cells to home to the epithelial compartments (mucosal immunity). In summary, we provide a paradigm for a vaccine approach that can be rapidly engineered and customized for any future pathogens that require induction of effective tissue-resident memory responses in epithelial tissues.

## 1. Introduction

Antigen presentation is an essential immune process that is responsible for T cell immune responses. Peter Charles Doherty and Rolf M Zinkernagel were the first to present the evidence that “the interaction of cytotoxic T cells with other somatic cells budding lymphocytic choriomeningitis (LCM) virus is restricted by H-2 gene complex” [1]. This discovery “of the specificity of the cell-mediated immune defense” had an immediate and widespread impact on immunological research. The Nobel Prize in Physiology or Medicine was awarded to Peter Charles Doherty and Rolf M Zinkernagel 20 years after their original discovery, recognizing their findings as one of the pillars of basic immunological principles. Major histocompatibility complex (MHC)-restricted T-cell recognition has initiated a considerable number of subsequent studies resulting in molecular understanding of T-cell recognition of virally infected target cells. Moreover, as addressed in the Zinkernagel Nature Lecture from December 1996, “recognition that peptides derived from viruses, bacteria or parasites are presented to T cells via MHC class I or class II molecules” suggested that instead of live, and therefore potentially harmful infectious agents, peptides could be used as vaccines to induce T-cell responses” [2]. In 1986, the first recombinant vaccine against hepatitis B virus (HBV) was approved by the FDA, which marked the beginning of the era of recombinant protein-based and peptide-based vaccines. However, subunit vaccines that consist primarily of peptides or proteins, in contrast to inactivated or attenuated pathogen or vector vaccines, can face significant limitations concerning immunogenicity, thus, they require carriers and adjuvants to counterbalance the low vaccine efficiency.

Here, we present a unique vaccine strategy based on the heat shock protein gp96, that provides antigen specificity and adjuvanticity within one molecule. Dr. Podack’s inspiring work on the development of an anti-tumor gp96 vaccine twenty years ago was successfully translated into numerous pre-clinical and clinical studies, including Phase 1b/2 clinical trial (NCT02439450) for patients with non-small cell lung cancer (NSCLC). Over the last decade, our lab has been extensively involved in the development of vaccines against human immunodeficiency virus/simian immunodeficiency virus (HIV/SIV), malaria, Zika virus (ZIKV), cytomegalovirus (CMV) and COVID-19 [3,4,5,6,7,8,9].

We propose that the recombinant protein gp96-Ig vaccine represents an “All for one and One for all” vaccine approach because different pathogens could be targeted by the gp96 vaccine, and gp96 could be engineered as a universal vaccine targeting numerous conserved antigens derived from pathogens or tumors, at the same time.

## 2. Gp96 as Molecular Chaperon

It was over 60 years ago that one of the students in Dr. Ritossa’s lab raised the temperature in the incubator in which he had some *Drosophila* salivary gland cells [10]. The next day, Dr. Ritossa discovered that heating induced puffs at the various regions of the polytene chromosomes. These puffs were areas of localized transcription that correlated with an increase in several families of heat shock proteins (HSPs). This extraordinary combination of serendipity, curiosity, and knowledge led to the identification of HSPs, a group of intracellular proteins that are induced when a cell undergoes several types of environmental stresses (heat, anoxia, glucose starvation) [10,11,12]. HSPs are ubiquitous, occurring in all organisms, from bacteria and yeast to humans, in various intracellular compartments [13]. Proteins within the HSPs family are categorized into families based on their molecular weights ranging from 10-150kDa (HSP27, HSP70, HSP90, HSP96) [13,14].

Gp96 (Grp94 or ERp99) is a member of the beta subfamily of HSPs 90 protein chaperon, a paralog of HSP90, found in the endoplasmic reticulum (ER) [15,16,17]. Under stress conditions (glucose starvation, oxidative stress, ER calcium depletion, and accumulation of misfolded proteins), gp96 is strongly upregulated and accelerates its function as a molecular chaperone [17,18,19]. Gp96 plays a critical role in folding proteins in the secretory pathway, such as Toll-like receptors [20], glycoprotein IX subunit [21], insulin-like growth factors (IGFs) [22,23], proinsulin [24], and integrins [25]. These interactions are associated with macrophage host defense, activation of platelets in blood clotting, cell growth, and cell adhesion [26]. Gp96 is also one of the major calcium-binding proteins in the ER regulating calcium homeostasis [26,27]. Gp96 protein consists of signal peptide (SP), N-terminal domain (NTD), C-terminal domain (CTD), middle domain (MD), and KDEL retention signal (Figure 1) [26,28,29]. NTD has ATPase activity and contains a nucleotide binding site and peptide binding site [30], while MD contains a catalytic loop for ATP hydrolysis [31]. In the CTD, there is a dimerization domain and client protein binding site [32,33]. Gp96 is a dimer that can change its state from closed to open depending on the ATP binding and ATP hydrolysis [26,28].

The peptide-binding channel in Gp96 supports to the proposal that Gp96 interacts with the peptide repertoire of the ER as part of the MHC peptide presentation machinery [16,34] and, therefore, has been implicated as an essential immune chaperone to regulate both innate and adaptive immunity.

## 3. Gp96 as an Immune Modulator

Once cells become infected by a pathogen or upon apoptotic stimuli or any other stressor, cells will eventually die, and the intracellular content, including HSPs, will become a part of the extracellular danger-associated molecular patterns (DAMPs) [35,36]. Extracellular gp96 will activate antigen-presenting cells (APCs) and will further induce both arms of immune responses, innate and adaptive immune responses (Figure 2).

Gp96 and other HSPs have been described as ideal cross-priming vehicles because of their property to bind intracellular and extracellular peptides and shuttle them between different antigen processing compartments inside the cell and between cells [38]. Pioneering work by Dr. Srivastava demonstrated that several HSPs (HSP7, gp96, and HSP90) can induce immunity against autologous tumor preparations from which these HSPs had been isolated [39,40]. Furthermore, it was well established that gp96 binds to several different receptors on APCs (Toll-like receptor (TLR) 2, TLR4, SRA, LOX1) [41,42,43] including the endocytic receptor, CD91 responsible for the internalization of gp96-peptide complex and simultaneous activation/maturation of APC (up-regulation of co-stimulatory molecules, pro-inflammatory cytokines, and NO release) and further activation of NK cells [44,45,46,47,48,49] (Figure 2). Endocytosed gp96 chaperoned peptides are presented within the MHC class I molecules and recognized by CD8 T cells. Specific activation of CD8 T cells is independent of the haplotype of the cells from which the HSP-peptide complexes originate [48,50,51]. The process of presentation of exogenous peptides within MHC class I molecules is known as cross-presentation, and gp96 was recognized as a very potent activator of the cross-presentation pathway in dendritic cells [48] (Figure 3). Cross-presentation pathways have two different pathways of delivering exogenous antigens to MHC class I molecules: vacuolar and endosome-to-cytosol pathways and gp96 engages in both [52] (Figure 3). In the endosome, gp96-peptide complex gets dissociated, and peptides escape to cytosol where they reach TAP which transports peptides to nascent MHC class I molecule in ER, and finally the MHC I-peptide complex is shuttled towards the cell surface [44,52,53] (Figure 3). However, some of the gp96-peptide complexes can also be processed in the recycling endosomes that carry MHC I molecule, this seems to be particularly important pathway for non-classical MHC I molecules, such an HLA-E [52,54,55,56] (Figure 3).

Immunogenic gp96 binds a variety of endogenous/exogenous peptides and engages conventional (cDC) and plasmacytoid (pDC) dendritic cells, monocytes/macrophages, and all other cells expressing CD91 [52,58]. In addition to the immunostimulatory properties of HSPs, it has been confirmed that HSPs also support an inhibitory role in the activation of autoimmunity, allograft rejection, and tumor immunity [59,60,61,62,63]. It has been reported that gp96 drives dichotomous T-cell response [58] depending on the dose: a low dose of gp96 will primarily engage with cDC and will result in the Th1 responses and expansion of CTLs, while a high dose of gp96 will activate pDC and prime CD4+T regulatory cells [60,61,62,63,64,65,66].

## 4. Recombinant Fusion Protein gp96-Ig

Eckhard R. Podack (1943–2015) was fascinated by the immune roles of gp96, and in the late 1990s, he developed a gp96-Ig-secreted vaccine approach [67]. To be immunogenic, gp96 must be outside of the cells, therefore Eckhard engineered a secreted form of gp96 simply by replacing the KDEL retention signal of gp96 with the Fc portion of the IgG1 molecule (Figure 1) [67]. In this way, gp96-Ig gets into the secretory pathway and is released from the cell in a complex with endogenous/exogenous peptides (Figure 1C).

At first, this engineered form of gp96-Ig was expressed in numerous tumor cell lines and tumor-secreted gp96-induced tumor-specific CD8 T cells that were required for tumor rejection [67]. Molecular and cellular gp96-Ig requirements for enhanced cross-antigen presentation to CD8 T cells have been described [47]. Secreted heat shock protein, gp96-Ig-chaperoned peptides enhance the efficiency of Ag cross-priming of CD8 cytotoxic T lymphocytes (CTL) by several million-fold over the cross-priming activity of unchaperoned protein alone [47]. Gp96 also acts as an adjuvant for cross-priming by unchaperoned proteins, but in this capacity, gp96 is 1000-fold less active than as a peptide chaperone [47]. Gp96-mediated cross-priming of CD8 T cells requires B7.1/2 co-stimulation but proceeds unimpeded in lymph node-deficient mice in the absence of NKT and CD4 cells and without CD40L [47]. Gp96-driven MHC I cross-priming of CD8 CTL in the absence of lymph nodes provides a novel mechanism for local, tissue-based CTL generation at the site of gp96 release. Podack and Strbo proposed that the gp96-Ig vaccine induced CD8 T cell responses and may constitute a critically important, early detection, and rapid response mechanism that is operative in parenchymal tissues for effective defense against tissue-damaging antigenic agents [37,47].

Srivastava and colleagues were the first to recognize the potential role of HSPs in the immune response to cancer [38,43]. They demonstrated that HSPs complexed with antigenic peptides obtained from lysed tumor cells (or virus-infected cells) provide an optimal source of tumor antigens, generating DC with improved cross-presentation capacity. Tumor-derived gp96 is taken up by APCs and subsequently cross-presented to stimulate potent CD8-mediated anti-tumor immunity [21,37,38,43,68] (Figure 3). To take advantage of this unique adjuvant effect and the ability to transport relevant peptides, we set up a model system that imitates necrotic cell death regarding the release of HSP (cell-secreted gp96-Ig). This system allowed us to analyze the immunological effects of HSP in vivo independent of tumor/infectious agents and cell death. We found that parenteral administration of gp96-Ig-secreting tumors triggers robust, antigen-specific CD8 cytotoxic T lymphocyte expansion combined with activation of the innate immune system [47,67,69]. Tumor-secreted gp96 causes the recruitment of dendritic cells (DCs) and natural killer (NK) cells to the site of gp96 secretion and mediates DC activation via binding to CD91 and Toll-like receptor-2 and Toll-like receptor-4 [47,49,70] (Figure 3). The endocytic uptake of gp96 and its chaperoned peptides triggers peptide cross-presentation via major histocompatibility complex (MHC) class I and strong, cognate CD8 activation independent of CD4 cells [37,47] (Figure 3). In this model system, CD8 CTL expansion can be precisely quantitated within 4 to 5 days of vaccination by the use of adoptively transferred, T-cell receptor (TCR) transgenic, green fluorescent protein (GFP)-marked CD8 T cells [47,49,70]. Using this test system, we showed that in our model system, tumor-induced immune suppression is antigen nonspecific and can be overcome by frequent immunization or by the absence of B cells [69].

In addition to the anti-tumor cell-based secreted gp96-Ig vaccine approach, which was successfully used in Phase 1b/2 clinical trial against non-small cell lung cancer (NSCLC) (NCT02439450 [71]), we have also generated an anti-infectious cell-based secreted gp96-Ig vaccine approach [37] (Figure 1). The major difference between these two platforms is that in the anti-tumor approach, we use tumor cells as a source of antigens by transfecting gp96-Ig construct in individual tumor cell lines, while in the anti-infectious approach, we have chosen the ATCC human embryonal kidney (HEK)-293 cell line, as a vehicle for gp96-Ig delivery in combination with different antigenic peptides that are co-transfected in the cell where the final product is gp96-Ig complexed with array of different antigenic peptides [37] (Figure 1C).

The reason why the secreted gp96-Ig vaccine approach is unique and has an advantage compared to other recombinant protein vaccine approaches is that gp96-Ig-induced antigen-specific CD8 T cells can migrate to mucosal surfaces and provide immediate and enhanced protection at the most likely entry site of invading pathogens [70] (Figure 4). We found that antigen-specific CD8 T cells are generated not only systemically (in blood and spleen) but also in the epithelia/mucosal compartments such as the gut, reproductive tract, and the lungs [3,4,7,8,9,37,70]. This was the main reason why we decided to adapt this anti-tumor vaccine approach to emerging infectious diseases. Expanding the analysis to mucosal compartments, we found a dramatic increase in simian immunodeficiency virus/human immunodeficiency virus (SIV/HIV)- specific CD8 T cells in the rectal mucosa of SIV vaccinated macaques (up to 30% of all CD8 T cells in the rectal mucosa are SIV-specific) (Figure 4) [3,4]. *Plasmodium falciparum* CSP and AMA1-positive CD8 T cells were found in the liver of malaria-vaccinated non-human primates (NHPs), while zika virus (ZIKV)- and cytomegalovirus (CMV)-specific CD8 T cells were found in the placenta and decidua of vaccinated mice and humanized mice, and finally, melanoma- and SARS-CoV-2- protein S- specific CD8 T cells were found in the lungs and bronchioalveolar lavage of gp96-Ig-melanoma and gp96-Ig COVID-19 vaccinated mice (Table 1 and Figure 4) [5,6,7,8,9].

## 5. How Does gp96 Induce Mucosal Immune Response?

We observed that gp96-Ig immunization increases the frequency of the subset of DC, CD11c^high^ MHC class II^high^ CD103^+^ cells at the site of vaccination [70] (Figure 4). Phenotypic analysis of CD11c^high^ MHC class II^high^ CD103^+^ cells revealed that these cells are also CD8^neg^ and express CD11b, CD40, CD80, CCR7, B220, and low levels of CD86 and Gr-1. CD103^+^ CD11b^+^ and CD103^+^ CD8^neg^ DC populations are more prominent in mesenteric lymph nodes and colonic lamina propria. Recently, CD8^neg^ CD11b^high^ MHC class II^high^ CD11c^high^ CD103^+^ DC population was found within the omental tissue of normal healthy mice. These cells cross-present exogenous antigen for MHC class I-restricted T-cell responses. In light of this finding and the findings of Bedoui et al. [72] that CD103^+^ DCs are the main migratory subtype with dominant cross-presenting ability, induction of CD103^+^ DCs by gp96 represents an ideal vaccination strategy for priming effective immunity. In addition, we have also confirmed that CD103^+^ DCs are responsible for the activation of CCR9^+^ CD8 T cells. Immigrating cells were clearly identifiable by the expression of high levels of CD44, CD103, CCR9, and α4β7 and downregulation of CD62L and CCR7 [70] that allow these cells to cross endothelial cells and migrate to the gut tissue. The use of this immunological mechanism seems ideally suited for vaccine purposes, including stimulation of cellular mucosal immunity against SIV/HIV (Figure 4).

Tissue-resident memory (TRM) T cells have been recognized as a distinct population of memory cells that are capable of rapidly responding to infection in the tissue without requiring priming in the lymph nodes [73,74,75,76]. In our recent work [7,9] we confirmed that gp96-Ig, secreted from allogeneic cells expressing full-length SARS-CoV-2 protein S and *Plasmodium falciparum* CSP and AMA1 proteins generates powerful, protein S polyepitope-specific CD4+ and CD8+ T cell responses in both lung interstitium and airways as well as CSP- and AMA-1-specific CD8+ T cells in the liver. Importantly, antigen-specific cells were characterized as tissue memory-specific cells (CXCR6+, CD69+, and CXCR3+) (Figure 4). These findings were further strengthened by the observation that protein-S -specific CD8+ T cells were induced in human leukocyte antigen HLA-A2.1 transgenic mice and *Plasmodium falciparum*-specific CD8 T cells in non-human primates, thus providing encouraging translational data that the vaccine is likely to work in humans. Therefore, vaccination strategies targeting the generation of TRM and their persistence may provide enhanced immunity compared with vaccines that rely on circulating responses [77].

The placenta acts as a barrier against infections due to multiple unique structural, cellular, and immune properties. The detrimental effects of congenital viruses, including ZIKV and HCMV, on pregnancy and fetal outcomes occur in part because of impaired placental function. Targeting the congenital infection with vaccines offers the best opportunity to generate protective immunity in the most vulnerable population of child-bearing women before viruses have a chance to cross the placental barrier and before infection takes hold in the fetus. We have succeeded in generating a vaccine that expresses gp96-Ig and ZIKV envelope protein and HCMV gB and pp65 protein. We confirmed that vaccination with secreted gp96-Ig is safe and induces CD8 T cell responses in maternal decidua (Figure 4) [5,6].

## 6. “All for One”

In the last decade, we developed the following emerging disease gp96-Ig vaccines, combining secreted gp96-Ig protein with the following antigens (Table 1 and Figure 4):

We have tested the immunogenicity of these vaccines in different animal models: mice, NHPs, and humanized mice; however, vaccine efficacy was only tested in an SIV infection model [3]. A combination of cell-secreted gp96-SIV-Ig and gp120 protein as immunogens achieved a significant reduction (73%) in the risk for SIV acquisition in NHPs [3]. Protection from mucosal SIVmac251 infection was associated with strong mucosal CTL and non-neutralizing antibody responses. The antibody isotype suggested Th1 polarization. Our control vaccines had no protective effect even though they generated either SIV-specific CTL or SIV-specific antibodies, suggesting that both SIV-specific CTL and antibodies are required for protection. 73% immunization efficiency is an encouraging starting point for further development of the SIV immunization strategy.

To this day, the cell-based secreted gp96-Ig vaccine approach was tested and completed in a Phase 1b/2 study of Viagenpumatucel-L (HS-110) in combination with multiple treatment regimens in patients with non-small cell lung cancer (NSCLC) (The “DURGA” Trial)(NCT02439450) [71,78]. HS-110 was well tolerated when administered in combination with nivolumab, an anti-programmed cell death receptor-1 (PD-1) monoclonal antibody. Significantly longer progression-free survival (PFS) and median overall survival (OS) in both checkpoint inhibitors (CPI) naïve and CPI progressor cohorts were observed [78]. In addition, a trend of improved OS in baseline blood tumor mutational burden (bTMB) patients in the CPI progressor cohort was also reported. Further clinical evaluation of HS-110 is warranted in both CPI naïve and CPI progressor NSCLC patients [78].

We are currently testing the efficacy of all other vaccines in different pre-clinical emerging disease models.

## 7. “One for All”

The recent pandemic highlighted the need to be prepared for the next emerging disease, in particular, the idea of having a vaccine that will give people a baseline level of immune memory to diverse pathogen strains with pandemic potential (flu, COVID variants) so that there will be far less disease and death after encountering a new emerging pathogen. Unlike current mono- or polyvalent vaccines, which confer immunity to one or several antigenic variants of a pathogen, universal vaccines do this by targeting an element of the pathogen that remains the same across all variants. Given the gp96 propensity to carry all peptides of a cell that will be selected in the recipient/vaccinee for MHC I loading, including transfected antigens, gp96 has the broadest, theoretically possible antigenic epitope-spectrum for cross-priming of CD8 T cells by any MHC I type. More so, it has been recently shown that tumor-derived gp96 can induce protection against lymphocytic choriomeningitis virus (LCMV) infection and against tumors, thereby emphasizing that gp96 fits in the “one for all” vaccine category [79,80]. Given the antigenic similarity between cancer and embryonic tissues [81] and the capability of gp96 to bind the entire peptide repertoire generated within the cell [82], there is a rational basis for the use of placenta-derived gp96 as a multivalent prophylactic cancer vaccine. [83,84]. In addition, gp96-activated DC can take up antigenic proteins and, after processing, present their epitopes via MHC II, thereby promoting antibody production by B cells [3,37]. Gp96 is thus a powerful Th1 adjuvant for CTL priming and for stimulation of Th1-type antibodies [3,8,37]. Current vaccines are all made against unique pathogen antigens rather than antigens that are shared by different pathogens. Universal preventative vaccines, such as recombinant gp96 vaccines, could activate the immune system based on the predicted changes in infected cells or transformed tumor cells and efficiently recognize and control emerging pathogens and tumors.

However, cell-based delivery of recombinant protein gp96-Ig by an irradiated human cell line (HEK-293) represents a major hurdle for further clinical development. Cell-based technology is impractical for large manufacturing, and the gp96-Ig vaccine requires cold-chain storage and distribution. Other types of vaccine delivery (recombinant DNA and RNA) are currently under investigation.

## 8. Conclusions

Secreted recombinant gp96-Ig vaccines act by initiating the innate immune response and activating APCs, thereby inducing a protective adaptive immune response to either a tumor or pathogen antigens. Gp96 is efficient in antigen cross-presentation at physiologic concentrations of antigen (picogram range), so endogenous amounts of gp96-antigenic peptide complexes secreted by vaccine cells are sufficient to induce robust memory CTL responses in epithelial tissues without the addition of other adjuvants. Furthermore, this vaccine platform is proof of concept for the prototype vaccine approach for all emerging infectious pathogens that require induction of effective tissue-resident memory responses in epithelial tissues.

## Figures and Tables

**Figure 1 cells-13-00072-f001:**
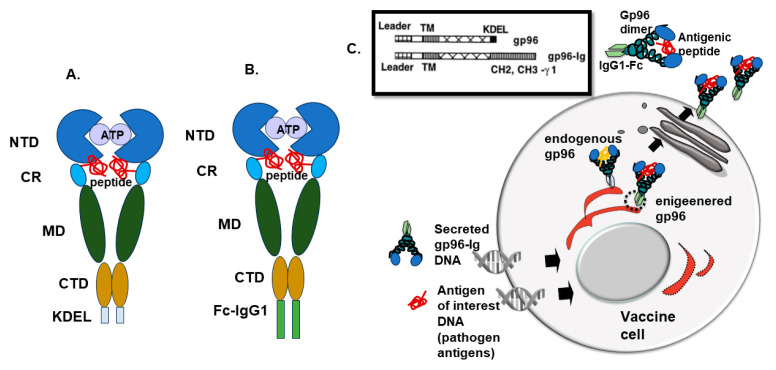
Schematic representation of gp96 and gp96-Ig structure and cellular distribution. N-terminal domain (NTD); the charged linker region (CR); the middle domain (MD). The C-terminal domain (CTD), and KDEL, the endoplasmic reticulum (ER) retention/retrieval ligand for the KDEL receptor (modified from [26]) (**A**) and Fc-IgG1 in gp96-Ig fusion protein (**B**). Gp96-Ig fusion protein co-expressed in the vaccine cells together with antigens of interest (pathogen antigens). Engineered gp96-Ig in complex with the transfected peptides is secreted outside the cells, while endogenous gp96 is retained in the ER (**C**).

**Figure 2 cells-13-00072-f002:**
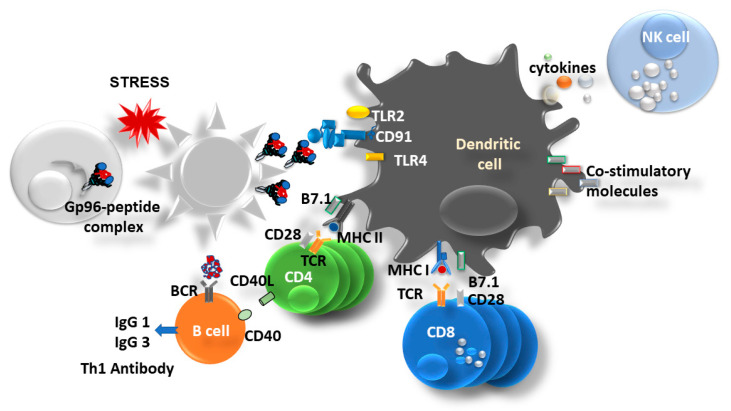
Gp96 is an immune modulator. Different environmental and endogenous stress signals will lead to cell death. Extracellular gp96 stimulates dendritic cells via different cell receptors (TLR2/TLR4, CD91), which results in the activation/maturation of dendritic cells (up-regulation of co-stimulatory molecules), secretion of pro-inflammatory cytokines (IL-12, IL-6), and activation of NK cells. At the same time, gp96 chaperoned peptides will be cross-presented to CD8 and CD4 T cells within MHC class I and MHC class II molecules, respectively. Activated CD4 T cells will further activate B cells that will produce Th1-type antibodies (modified from [37]).

**Figure 3 cells-13-00072-f003:**
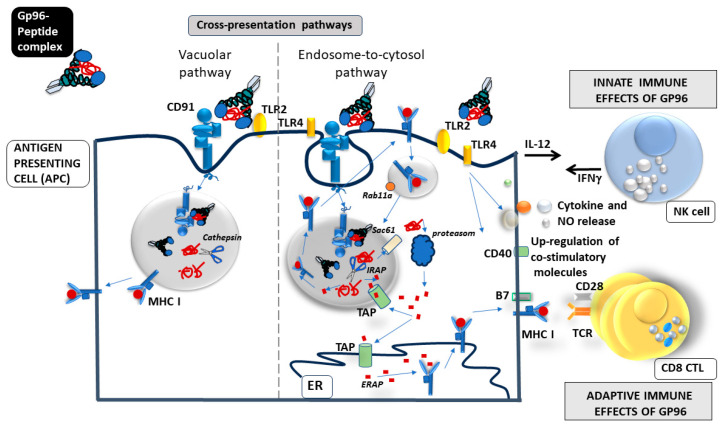
Schematic overview of gp96-Ig-peptide/antigen complex endocytosis and antigen cross-presentation. Internalized gp96-Ig chaperoned peptides/antigens can be presented via two cross-presentation pathways: vacuolar or endosome-to-cytosol pathway. In the vacuolar pathway, peptides/antigens are degraded in endosomes by cathepsin S and subsequently loaded onto major histocompatibility complex (MHC) I. In the endosome-to-cytosol pathway, peptides/antigens are transported into the cytosol for proteasomal degradation via Sec61 translocon. Afterward, antigen-derived peptides are transported back into the endosomes or the ER via a Transporter associated with antigen processing (TAP). There, they are trimmed by Insulin responsive aminopeptidase (IRAP) (endosomes) or Endoplasmic reticulum (ER) aminopeptidases (ERAP) and loaded onto MHC I. Recycling MHC I (Rab11a+ endosomes) are loaded with peptides within the endosomes. Gp96-Ig activates APCs through upregulation of co-stimulatory molecules (B71/2 and CD40) and secretion of pro-inflammatory cytokines (IL-12, IL-6) that activate NK cells (activation and IFN gamma production) (innate immune effects) while peptides presented within MHC I molecules are recognized by antigen-specific CD8+ T cells and further induce their activation/proliferation (adaptive immune effects). (Modified from [57]).

**Figure 4 cells-13-00072-f004:**
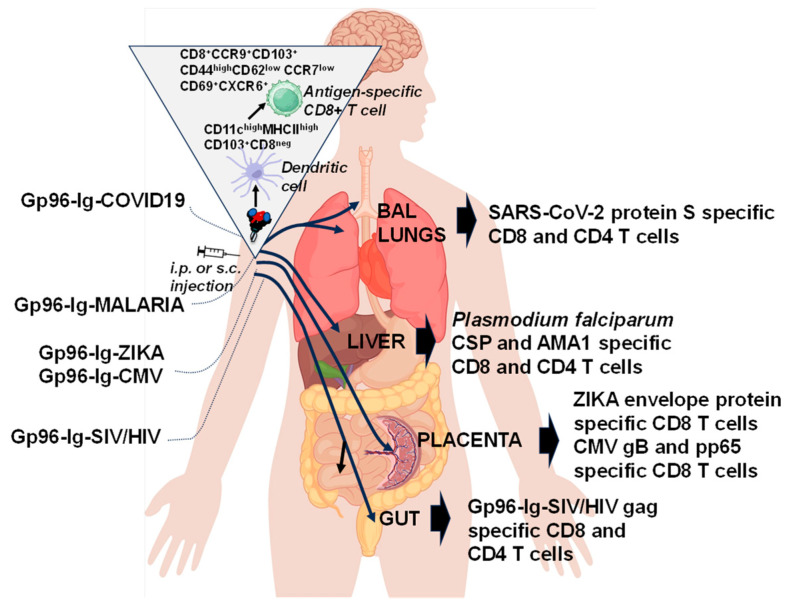
Proposed pathway for antigen-specific CD8+ T cells induction in different epithelial compartments by secreted gp96-Ig vaccine (based on the pre-clinical models). Recombinant gp96-Ig delivered by subcutaneous (s.c.) or intraperitoneal (i.p.) route of vaccination will induce activation/migration of CD103+ DCs that are crucial for induction of CD8+ cells expressing CCR9 and CXCR6 that will allow these cells to cross endothelial cells and migrate to the gut, reproductive (placenta) and lung tissue and airways.

**Table 1 cells-13-00072-t001:** Secreted gp96-Ig vaccines against different emerging pathogens.

Vaccine	Gp96-Ig Concentration *	Co-Expressed Antigens (Full-Length Proteins)	Vaccine Induced Immune Responses
SIV/HIV vaccine	1 μg/mL	SIV/HIV gag	Blood
		SIV/HIV ReTaNef	Rectal and vaginal lamina propria
		SIV/HIV gp120	Rectal and Vaginal Epithelial [3,4]
Malaria vaccine	2 μg/mL	*Plasmodium falciparum* CSP	Blood Liver [9]
		*Plasmodium falciparum* AMA1	
Zika vaccine	0.5 μg/mL	Zika Pre-membrane Envelop	Blood Spleen Placenta
			Decidua [5,6]


CMV vaccine	0.5–1 μg/mL	Human cytomegalo virus (HCMV) gBpp65	Blood Spleen Placenta Decidua [6]
	1 μg/mL	SARS-CoV-2- protein S	Blood Spleen Lungs Air ducts [7,8]
COVID-19			

* Gp96-Ig concentration measured by ELISA (1 million vaccine cells/1 mL media after 24 h hours culture).

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
