# Peer review of "“All for One and One for All”: The Secreted Heat Shock Protein gp96-Ig Based Vaccines"

_cells, 2023, doi:10.3390/cells13010072_

Round 1

Reviewer 1 Report

Comments and Suggestions for Authors

The authors present a review on the interesting gp96-Ig-based vaccine strategy, historically promoted by the team under the direction of Dr. Podack. Thus, they update their previous reviews on the subject, the last one published in 2013 (Strbo et al., Immunol Res, doi: 10.1007/s12026-013-8468-x), in a post-COVID scenario. The review is pertinent, offering the reader the approach and its advantages to promote immunity of co-formulated antigens at the cellular and mucosal levels. It generally reads easily for someone outside the Immunology field, which is my case. I understand that the focus is set on this discrete vaccine approach, which the authors have proven promising in pre-clinical stages, but I miss some deeper comparison to alternative approaches and a more critical view, beyond praising the excellence of this setting. Before acceptance, the spelling and style must be thoroughly checked. English is not my mother language, but I find hard to follow and understand many sentences due to the lack of articles of misuse of hyphens. Some general and particular advice for improvement follows.

Major points

Care should be taken on the spelling all along the manuscript. Some corrections are suggested below, but a thorough revision of style should be made.

Is the statement in lines 211-214 right or an overstatement? “compared to ALL other vaccine approaches”. Does this include live attenuated vaccines given orally or inhaled?

I see a problem with Fig. 4. I think it is misleading. All results presented here are pre-clinical, but a human body is depicted. If this is a hypothetical translation of animal studies to the clinics, it should be clearly stated that it is meant so. Though it can be predicted on this basis, no evidence supports, as intended here, these preclinic results in humans. Primates or mice, as corresponds, should be represented.

The ”One for all” Chapter, #7, is too optimistic. The idea of universal vaccines based in this system is attractive, but this point should be better explained, and putative limitations discussed. Reading this passage one may think that classic vaccine strategies, specifically devised to prevent a particular tumor or infection are not worth. The value of recombinant gp96-vaccines cross-protection should be presented in its proper context. Please check that Finn’s lab reference 80 is indeed appropriate for the flu challenge mentioned. This particular reference deals with LCMV infection. Check references for accurate matches to the statements in the text. What does “not healthy liver” mean here?

Minor suggestions

The abstract presents the review in the context of the Doherty & Zinkernagel 50 year tribute. In lines 15-16 it is stressed “our” vaccine approached/strategy. The authors absolutely have the right to claim this, but I do not think that this is the appropriate language in an abstract, or even throughout the paper. The reader should conclude this from the authorship of the cited work. It would be more elegant as “(…), including heat shock protein-based vaccines. In this review we highlight (…) of a promising strategy grounded on the use of recombinant gp96, developed by our lab over the last two decades:” Note that this contrasts with other passages of the article in which the author’s work is referred to in the third person (line 175: Podack and Strbo proposed…)

The last paragraph of the Introduction about the rainbow eucalyptus tree, although beautifully metaphorical, is a bit exaggerated for my personal taste on a scientific document. I leave this decision to the authors, but I would remove the metaphor and keep it for dissemination purposes outside the environment of a scientific journal.

Please respect scientific nomenclature. Drosophila and Plasmodium falciparum should be in italics, genus name starting with a capital letter. SARS-CoV-2 instead of SARS-COV-2. COVID-19, instead of COVID19, cytomegalovirus instead of cytomegalo virus. ‘Gag’ instead og ‘gag’ when referring to the protein…

Importantly, in Chapter 7, please do not write "COVID strains", but variants. Antigenicity variation does not define strains, but variants, or types and subtypes. In line 301, please write “antigenic variants of a pathogen” instead of “strains of a disease”.

Comments on the Quality of English Language

Sentences missing articles or lacking full sense (the whole manuscript should be reviewed, as the lack of articles may mislead the reader). See these examples:

Abstract. “Two scientists shared…” Not ‘any’ two scientists. These two scientists!

Lines 69, 129: ‘the’ endoplasmic reticulum

The paragraph in lines 202-210 is missing several articles, making it a bit hard to follow.

Line 218: ‘expanding analysis’ should be ‘expanding analyses’ or, rather, ‘expanding the analysis’.

Line 219: we found ‘a’ dramatic increase…

Line 244: represents ‘an’ ideal…

Lines 276, 277: ‘the’ following

Line 300: encountering ‘a’ new emerging pathogen

Line 316: ‘than on antigens’ remove the article here, but use it in ‘could activate THE immune system’.

Please improve writing (some examples follow):

Line 58: the incubator ‘that’ he had some… The incubator ‘in which’ he had some…

Line 97. (…) content of all intracellular proteins (…). Is it meant “all intracellular proteins” or “the intracellular content”?

Lines 163-165: This sentence makes no sense at all. Something must be missing.

Please use abbreviations instead of full names when previously described:

Line 97-98: HSPs

Line 308: tumors instead of tumor?

Please match verbal tenses:

Lines 98-99: (…) gp96 WILL ACTIVATE (…) and further INDUCES (…)

Line 303: Given the gp96 property to carries all peptides (???)

Please use plurals correctly, also in abbreviations:

Line 99, 180 (APCs), see also within Fig.3

Lines 179, 180: HSPs

Line 43 Proteins or protein?

Line 207, cells line should be cell lines?

Line 279: media or medium?

Please use hyphens properly (only some examples follow):

Line 175: “gp96 based vaccines induced CD8 T cell responses” should be “gp96-based vaccine-induced CD8 T cell responses”

Line 183: Tumor-derived

Line 186: cell-secreted

Lines 291-292: SIV-specific

Figure 1 is not cited in the text. It would be useful to make reference to it along the text.

Fig. 1C. The lettering in the squared area top left maybe too small to be read when printed!

Line 68: “is one of the HSPs 90 kDa beta member” (?) “is a member of the beta subfamily of HSP90 protein chaperons”?

In Fig. 3 IFNg should be IFN-g

Line 203, 222. Please explain to a naïve reader NSCLC, NHP, SIV....

Line 205: these, not this

Line 208: delete ‘cells’ (redundant)

Line 217: Is “apply” really meant? Adapted?

Chapter 5. Please check the space between the CDs and their superindexes. I think there should be no space between CD11c and high, etc, etc. Sometimes there is, sometimes there is not. Please homogenize.

Line 270: viruses instead of virus?

Titel of Chapter 6: Is there something missing?

Table 1. Please remove capital letters when inappropriate in last column.

What does (Ref) mean? Delete.

Is the tag “production” in the second column appropriate? Although explained in the foot note, production may not be an accurate term for concentration data.

Lines 281 to 284. This is iterative. It has been explained before, shown in Fig. 4 and in the Table.

Lines 291-293: “controls vaccines” should be control vaccines. “are required” instead of “are require”. I do not think I understand this sentence. You mean that BOTH CTLs and IgGs are efficient, but any of them individually are not? If so, it should be rephrased.

Line 301. Like current ‘mono- or polyvalent’ vaccines,…

Line 306: ‘it’ has been recently shown, instead of “recently has been shown”

Line 323: “immune response to either tumor or pathogen antigens” better than “immune response to tumor and pathogen antigens”.

Line 325. “secreted by vaccine cells”. This is misleading, please explain better.

Line 340 “in the “ is repeated.

Author Response

We would like to thank you for your constructive critique and suggestions how to improve our manuscript. Thank you!

We have thoroughly revised manuscript (all corrections are highlighted in the yellow color in the manuscript text). Here, we are going to discuss just the major and minor points.

Major points

The whole manuscript underwent extensive English revisions through University provided English writing revise editing service. 

  1. We agreed that statement "compared to all other vaccine approaches" is an overstatement. This is a revised sentence "compared to other recombinant protein vaccine approaches."
  2. We revised Fig 4 and changed the title to address the hypothetical translation of pre-clinical (animal) studies to the clinic " Proposed pathway for antigen-specific CD8+ T cells induction in different epithelial compartments by secreted gp96-Ig vaccine (based on the pre-clinical models). "
  3. As suggested, we included discussion about the putative limitations of gp96-Ig vaccine approach "However, cell-based delivery of recombinant protein gp96-Ig by an irradiated human cell line (HEK-293) represents a major hurdle for further clinical development. Cell-based technology is impractical for large manufacturing and the gp96-Ig vaccine requires cold chain storage and distribution. Other types of vaccine delivery (recombinant DNA and RNA) are currently under investigation" We have also corrected the statement regarding the Finn's lab Ref about LCMV challenge study and add new Ref that reflects the effect of pre-tumoral anti flu immune responses on the tumor outcome (new Ref 79). ".... protection against lymphocytic choriomeningitis virus (LCMV) infection and against tumors, thereby emphasizing that gp96 fits in the “one for all” vaccine category [78, 79].
  4. Minor suggestions
  5. As suggested "our vaccine approach" in the Abstract was replaced with the suggested "including promising strategies grounded on the use of recombinant gp96-Ig developed by our lab over the last two decades. "
  6. The last paragraph in the introduction was replaced with the following paragraph "Here we present a unique vaccine strategy based on the heat shock protein, gp96 that provides antigen specificity and adjuvanticity within one molecule. Dr. Podack’s inspiring work on the development of an anti-tumor gp96 vaccine twenty years ago, was successfully translated into numerous pre-clinical and clinical studies including Phase 1b/2 clinical trial (NCT02439450) for patients with non-small cell lung cancer (NSCLC). Over the last decade, our lab has been extensively involved in the development of vaccines against human immunodeficiency virus/ simian immunodeficiency virus (HIV/SIV), malaria, Zika virus (ZIKV), cytomegalovirus (CMV) and COVID-19 [3-9]"
  7. Scientific nomenclature throughout the manuscript was revised. 
  8. COVID strains was replaced with "COVID variants." 
  9. We would like to express our special gratitude for all the comments about the quality of the English language. All reviewer's suggested corrections are included in the revised version (highlighted in the yellow). In addition, as we have already mentioned, the whole manuscript underwent extensive English revision through university provided English writing revise editing service. 

Reviewer 2 Report

Comments and Suggestions for Authors

The manuscript “’All for one and One for all’: the Secreted Heat Shock Protein gp96-Ig Based Vaccines” discusses the immune activation effects of the gp96-Ig based vaccines in cancer and infectious diseases. Importantly, gp96-Ig based vaccines can provide antigen specificity and adjuvanticity at the same time and represent an “All for one and One for all” vaccine approach. While the authors have done a great job, there are some areas of the review require clarifications and complements.

1.      In abstract and conclusion section, some conclusive sentences should be integrated to highlight the meaning of your review and this vaccine strategy.

2.      In introduction section, the language of line 47~52 should be more academic and concise. The concept and history of gp96-Ig, which is detailed described in section 4, should be introduced briefly here.

3.      The authors should check the abbreviations in the review. For example, the authors used “MHC” it in Line31 and 35 but gave the full name of it in Line 92.

4.      In section 6, the authors should include a comprehensive summary of the current clinical trials involving gp96-Ig vaccines across diverse diseases. This will provide readers with a clearer understanding of the current status and potential clinical applications of gp96-Ig based vaccines.

5.      In order to comprehensive introduce the vaccine strategy, the limitations of gp96-Ig based vaccines should be discussed. This will provide directions for future researches.

Comments on the Quality of English Language

The quality of English language is overall satisfying. Some abbreviations should be checked.

Author Response

Thank you very much for your comments and suggestions.

  1. We have included conclusive sentence in the Abstract and conclusion Section. All changes in the text were highlighted in the yellow. "In summary, we provide a paradigm for a vaccine approach that can be rapidly engineered and customized for any future pathogens that require induction of effective tissue-resident memory responses in epithelial tissues."
  2. We agreed that language in the Introduction paragraph Ln 47-52 wasn't appropriate for scientific writing, and we replaced it with the brief concept and the history of gp96 vaccine. "Here we present a unique vaccine strategy based on the heat shock protein, gp96 that provides antigen specificity and adjuvanticity within one molecule. Dr. Podack’s inspiring work on the development of an anti-tumor gp96 vaccine twenty years ago, was successfully translated into numerous pre-clinical and clinical studies including Phase I and Phase II clinical trials (NCT02439450) for patients with non-small cell lung cancer (NSCLC). Over the last decade, our lab has been extensively involved in the development of vaccines against human immunodeficiency virus/ simian immunodeficiency virus (HIV/SIV), malaria, Zika virus (ZIKV), cytomegalovirus (CMV) and COVID-19 [3-9]"
  3. All abbreviation throughout the whole text were corrected.
  4. We included summary of the past clinical trials involving gp96-Ig vaccines against tumors. Gp96-Ig vaccines against emerging infectious diseases are still in the pre-clinical phase of testing. 

    "To this day, cell-based secreted gp96-Ig vaccine approach was tested and completed in a Phase 1b/2 study of Viagenpumatucel-L (HS-110) in combination with multiple treatment regimens in patients with non-small cell lung cancer (NSCLC) (The "DURGA" Trial)(NCT02439450)[70, 77]. HS-110 was well tolerated when administered in combination with nivolumab, an anti-programmed cell death receptor-1 (PD-1) monoclonal antibody. Significantly longer progression-free survival (PFS) and median overall survival (OS) in both check-point inhibitors (CPI) naïve and CPI progressor cohorts was observed [77]. In addition, a trend of improved OS in baseline blood tumor mutational burden (bTMB) patients in the CPI progressor cohort was also reported. Further clinical evaluation of HS-110 is warranted in both CPI naïve and CPI progressor NSCLC patients [77].

    We are currently testing the efficacy of all other vaccines in different pre-clinical emerging disease models."

  5. Limitations of gp96 vaccine strategy were discussed in the paragraph 7:

    "However, cell-based delivery of recombinant protein gp96-Ig by an irradiated human cell line (HEK-293) represents a major hurdle for further clinical development. Cell-based technology is impractical for large manufacturing and the gp96-Ig vaccine requires cold chain storage and distribution. Other types of vaccine delivery (recombinant DNA and RNA) are currently under investigation. "

Reviewer 3 Report

Comments and Suggestions for Authors

The manuscript on the gp96 protein is very helpful for understanding the adaptation of the HSP molecules to vaccination in humans. We are interested in the theme of the research.

Minor comments

The author should delete the title of Figure 4 in the panel.

We found several inconsistencies in the abbreviations of the manuscript, such as TLR and MHC.  

Author Response

Thank you very much for your comments!

We have revised the manuscript according to your suggestions and deleted Title in the Fig 4 as well as included abbreviations for TLR and MHC and corrected all other inconsistency regarding other abbreviations through the text. 

Best Regards,

Natasa